# Nutritional Composition of Breakfast in Children and Adolescents with and without Celiac Disease in Spain—Role of Gluten-Free Commercial Products

**DOI:** 10.3390/nu15102368

**Published:** 2023-05-18

**Authors:** Natalia Úbeda, María Purificación González, María Achón, Ángela García-González, Catalina Ballestero-Fernández, Violeta Fajardo, Elena Alonso-Aperte

**Affiliations:** Departamento de Ciencias Farmacéuticas y de la Salud, Facultad de Farmacia, Universidad San Pablo-CEU, CEU Universities, Urbanización Montepríncipe, Boadilla del Monte, 28660 Madrid, Spain; nubeda@ceu.es (N.Ú.); achontu@ceu.es (M.A.); angargon@ceu.es (Á.G.-G.); catalinaballestero@gmail.com (C.B.-F.); violeta.fajardomartin@ceu.es (V.F.); eaperte@ceu.es (E.A.-A.)

**Keywords:** breakfast, children, adolescents, gluten-free

## Abstract

Eating a nutritionally balanced breakfast can be a challenge when following a gluten-free diet (GFD). We assessed the ingredients and nutrient composition of 364 gluten-free breakfast products (GFPs) and 348 gluten-containing counterparts (GCCs), and we analysed the nutritional quality of breakfast in a group of Spanish children and adolescents with celiac disease (CD) (*n* = 70), as compared to controls (*n* = 67). Food intakes were estimated using three 24 h dietary records. The composition of GFPs and GCCs was retrieved from the package labels of commercially available products. Most participants (98.5%) ate breakfast daily, and only one person in each group skipped breakfast once. The breakfast contribution of the total daily energy was 19% in participants with CD and 20% in controls. CD patients managed a balanced breakfast in terms of energy (54% from carbohydrates; 12% from proteins; 34% from lipids) and key food groups (cereals, dairy, fruits), but their intake of fruits needs improvement. Compared to controls, breakfast in the CD group provided less protein and saturated fat, a similar amount of carbohydrates and fibre, and more salt. Fibre is frequently added to GFPs, but these contain less protein because of the flours used in formulation. Gluten-free bread contains more fat and is more saturated than is GCC. Sugars, sweets, and confectionery contribute more to energy and nutrient intakes in participants with CD, while grain products do so in controls. Overall, breakfast on a GFD can be adequate, but can be improved by GFPs reformulation and a lower consumption of processed foods.

## 1. Introduction

Breakfast is identified as a significant contributor to a healthy lifestyle and represents an important source of key nutrients in the diet of children and adolescents [1]. However, there is still no unanimous consensus on whether breakfast is the most important meal of the day [2,3,4]. Spanish dietary recommendations suggest that a healthy and a nutrient density-adequate breakfast should contribute around 20–25% of the total daily energy intake and it should constitute the triad: (1) dairy products (a glass of milk, fresh yogurt, or a portion of cheese); (2) cereals (bread, cookies, homemade pastries, or breakfast cereals); and (3) fruit or natural juice. Furthermore, it could also be complemented on some occasions by other protein foods, such as eggs, ham, nuts, etc. [2,5].

According to results by Ruiz et al. [2] from the Anthropometry, Intake and Energy Balance (ANIBES) Study in Spain, children consume breakfast frequently (93.4%); however, the highest prevalence of irregular and non-breakfast consumers were clearly identified among adolescents (12.3% and 7.6%, respectively). Breakfast contributed between 17 and 18% of the total daily energy intake in these population groups. The most consumed breakfast food was chocolate (mainly as chocolate-flavoured milk and powder), followed by baked goods and pastries, whole milk, and semi-skimmed milk. Recently, similar results were found by Cuadrado-Soto et al. [1] from the National Dietary Survey on the Child and Adolescent Population (ENALIA) Study, also conducted in Spain. According to these studies, Spanish youth are not meeting recommendations for breakfast, a fact that poses the intriguing question of what may be happening when there is a food restriction in cereal consumption, as in the case of people with celiac disease (CD). Wheat and other gluten-containing cereals are very common in the Spanish diet, so taking a complete and nutritionally adequate breakfast can be a real challenge for children and adolescents with CD. If unmastered, the selection of suitable gluten-free foods for breakfast is difficult and expensive for celiacs, and may turn boring and exhausting, causing youngsters to skip breakfast.

Skipping breakfast is more frequent among women, later adolescents, those living in single-parent households, and in lower socioeconomic positions [6]. Breakfast skipping has also been found to be positively correlated with overweight/obesity (OW/OB) and biomarkers of metabolic diseases [6,7,8,9,10]. Particularly, the Food, Physical Activity, Child development and Obesity (ALADINO) and the Healthy Lifestyle in Europe by Nutrition in Adolescence (HELENA) Studies, carried out on Spanish children, confirm the association between not eating breakfast daily and a higher prevalence of OW/OB [11,12,13]. However, these results should be interpreted carefully since there are other studies which show contrary results [7,14].

The limited evidence from longitudinal studies among children/adolescents suggests that skipping breakfast is also related with higher fasting insulin levels and that daily breakfast practice is linked to a significantly lower homeostasis model assessment—insulin resistance (HOMA-IR) index [4,15]. Moreover, cognitive function could also be affected by eating an adequate breakfast. Its regular consumption was similarly correlated with better academic performance scores [16,17].

Many studies also confirm the impact of a healthy breakfast, showing higher daily nutrient intakes, an improved daily total nutrient intake, a better compliance with nutritional recommendations, and a better overall diet quality [18,19]. Specifically, children and adolescents who eat breakfast on a regular basis, compared to those who do not eat breakfast, consume higher amounts of energy, dietary fibre, fruits, and vegetables, and fewer sugar-sweetened beverages [4]. In this context, those who eat a daily breakfast consisting of dairy products, breakfast cereals, and fruits have higher daily intakes of some critical micronutrients for their age group (calcium, iron, potassium, magnesium, zinc, and iodine) compared to breakfast skippers [4].

To our knowledge, there are no studies assessing breakfast in a population with CD. A strict and lifelong adherence to a gluten-free diet (GFD) is the first-line treatment and, currently, is the only effective therapy for patients with CD and all other gluten-related disorders, such as non-celiac gluten sensitivity or wheat allergy [20].

CD is a major public health problem worldwide, with the following global prevalence data depending on the diagnostic method employed: 1.4% based on serologic tests and 0.7% according to biopsy [21,22]. The prevalence of CD varies with sex, age, and geographic region. Particularly, different studies show that the incidence rates of CD in children are significantly higher (0.9% vs. 0.5%) than in adults [21,23,24,25]. In Spain, current CD prevalence in children could be much greater than that monitored in other European countries [24]. Diagnoses of the early stages of CD and the life-long exclusion of gluten are the main therapeutic approaches to the disease, which is multisystemic and affects multiple organs. Subjects with CD are more likely to have digestive problems because gluten triggers an immune response in the small intestine that impacts the mucosa and lowers the ability to absorb nutrients in the body. In children and adolescents with CD, malabsorption can cause growth and developmental problems such us weight loss, anaemia, irritability, short stature, delayed puberty, tooth enamel defects, neurological symptoms, including attention-deficit/hyperactivity disorder, learning disabilities, headaches, chronic fatigue and, over time, osteoporosis [26].

Due to the limitations of a GFD, children and adolescents particularly consume many processed products made specifically for them [27]. According to some authors, abusive use of these products can have long-term consequences, including systemic inflammation or intestinal microbiota alteration, that appear to contribute to the persistence of nutritional deficiencies [28] and cardiometabolic-related pathologies, such as obesity [29,30] or cardiovascular disease [31]. Because of cultural dietary habits and food recommendations for breakfast, breakfast is the meal of the day in which gluten-free processed foods are more likely to be introduced in Spain.

Taken together, all the aforementioned studies warrant the importance of studying breakfast habits to prevent serious health issues. There is very limited information on children and adolescents with CD, especially with regard to their breakfast diet. Therefore, the present study firstly aimed to assess the nutritional quality (based on ingredients and nutrients) of processed cereal-based products commonly consumed at breakfast, e.g., breads, breakfast cereals, bakery products, etc., both gluten-free (GFPs) and their gluten-containing counterparts (GCCs). Furthermore, the second objective was to analyse the breakfast quality (all foods included) of a group of Spanish children and adolescents with CD compared to a group of similar age and gender characteristics without the disease (control). This analysis was based on the evaluation of the consumption of the different food groups recommended by the Spanish Society of Community Nutrition (Sociedad Española de Nutrición Comunitaria, SENC) [5] (dairy products, cereals, fruits, etc.) as well as the quality of their nutritional composition. Processed foods are generally recognised as a source of high energy, saturated fats, trans-fatty acids, sugar, and salt. An excessive intake of these nutrients is perceived as the main risk reason for developing some of the major public health problems such as OW/OB, type II diabetes, cancer, and cardiovascular diseases [32]. Results should be valuable for nutritional education and food reformulation, especially when developing strategies to improve nutritional quality and reduce the consumption of processed GFPs for breakfast.

## 2. Participants and Methods

### 2.1. Participants

Current dietary data were obtained in a cross-sectional survey in children and adolescents diagnosed with celiac disease (CD) and healthy controls. The Celiac and Gluten Sensitive Association (Asociación de Celiacos y Sensibles al Gluten de Madrid, Spain) helped in the recruitment of the participants. The eligibility criteria for the CD group included ages between 4 and 18 years old, having a certified diagnosis of CD, being on a gluten-free diet (GFD) for more than a year, not consuming pharmacological supplements, and not being affected by digestive discomfort at the time of dietary assessment. Adherence to the GFD was tested in blood samples from all participants through the analysis of immunoglobulin A (IgA) antitissue transglutaminase antibodies (IgA-tTG). The control group (healthy) participants were enlisted from the general population when meeting the following inclusion criteria: healthy status (absence of diagnosed chronic disease); not having symptoms or signs of any digestive disease; and not taking pharmacological or nutritional supplements.

All subjects and guardians or caregivers were informed and asked for a written consent to participate before enrolling. The study was conducted following the legal requirements and guidelines for good clinical practice, as well as the World Medical Association Declaration of Helsinki on Ethical Principles for Medical Research involving Human Subjects (revised in October 2008). The procedure was authorised by the Ethics Committee for Human Studies in Universidad San Pablo-CEU (Authorization number 102–15).

### 2.2. Ingredients and Nutrient Content of Breakfast Products

The gluten-free products (GFPs) composition database, developed by our research team and available at the Universidad San Pablo-CEU institutional repository [33,34], was used for GFPs composition data. This food database was compiled using the nutritional composition and ingredient list data from labels, as previously described [34]. Gluten-containing counterparts (GCCs) were chosen from retail stores and were matched to GFPs based on the same product (equal name and presentation) and greatest similarity in ingredient list. Ingredient and nutrient data from GCCs were also collected from the labelling on their packaging. Nutritional compositional of GFPs, currently available on the market, was evaluated in contrast to their GCCs.

Ingredients were chosen according to their impact on the nutritional profile of GFPs and GCCs and because of their critical effects on human health (starchy ingredients, fats, sugars, and fibre). In particular, the top ten most frequently used ingredients were considered. To analyse the frequency of use of these critical ingredients in the formulation of GFPs and GCCs, the breakfast products were organised in the following groups: bread and similar; breakfast cereals, biscuits, sweets, and semi-sweets, pastries and cakes, and churros (a traditional Spanish breakfast food consisting of a deep-fried dough made up of wheat and modelled in long tubes).

### 2.3. Food Habits and Nutrient Intakes

Firstly, a trained dietitian collected diverse information from the participants (personal data, family history of disease, and medication) during a face-to-face interview. According to the recommendations of the European Food Safety Authority [35], an individual’s diet was estimated by applying three 24 h dietary records. The dietitian completed the first record with the assistance of the volunteers’ relatives when it was necessary. The other two 24 h dietary records were fulfilled via phone call with a time difference interval of one month. A Sunday or a holiday was recalled for one of the three 24 h dietary records. GFPs brands were registered and the composition of all GFPs consumed was included in the database of the software used for analysis. As we previously explained [36], labels do not record data on micronutrient composition; therefore, data on micronutrient intake from these products were not quantified.

The assessment of energy and nutrient intakes was carried out using the DIAL^®^ software, version 3.15 (Alce Ingeniería, Madrid, Spain) [37].

### 2.4. Statistical Analysis

For data analysis, IBM SPSS^®^ Statistics for Windows (version 27.0, Somers, NY, USA, 2021) was used. A Kolmogorov–Smirnov test was applied to confirm the normality of the target variables. Results are shown as mean ± standard deviation. Mean differences between GFPs and GCCs were assessed using the Student’s *t*-test. The analysis of categorical variables (descriptive data on ingredients) was handled using chi-squared test, and data are reported as frequencies (number of foods including a specific ingredient) and percentages (based on the total products within the group). Statistical significance was regarded only when *p*-values were lower than 0.05.

## 3. Results

A total of 70 participants with celiac disease (CD) (50% females and 80% children) and 67 non-celiac (control) (39% females; 69% children) took part in the survey. Mean age was 10.1 ± 3.7 for participants with CD and 10.3 ± 3.5 for controls. Most participants (98.5%) consumed breakfast every day, and only one person in each group skipped breakfast once.

### 3.1. Ingredients Used in Gluten-Free and Gluten-Containing Breakfast Products

A total of 364 GFPs and 348 GCCs were evaluated for ingredient and nutrient composition. Table 1 and Table 2 display the type of flours and starches used as ingredients in the formulation of GFPs and GCCs commonly consumed at breakfast by Spanish children and adolescents. 

As expected, GFPs are made with gluten-free flours such as rice, maize, pseudocereal, and legume flours; however, GCCs are mainly composed of wheat, rye, oat, and barley flours, regardless of the product group. Corn and rice cereal flours are the most frequently used, followed by legume flours in GFPs, whereas they are rarely used in GCCs. Notably, the only whole meal flour used is wheat flour, and it is only found in GCCs such as bread and similar, breakfast cereals, biscuits, sweets, and semi-sweet products. Regarding starches, a statistically higher frequency of use is observed among GFPs (Table 2). Gluten-free bread and similar, biscuits, sweets, semi-sweets, pastries, and cakes mainly include starch from corn, rice, potato, and tapioca. Only corn and wheat starch are used in GCCs.

Table 3 includes fat ingredients used in the formulation of GFPs and GCCs. Sunflower oil is the most frequently used ingredient in all breakfast products studied, and gluten-free breads include this oil more frequently, compared to GCCs. The least often used fats are palm, cocoa, coconut, butter, and cream in all groups of breakfast products, with hardly any significant differences between GFPs and GCCs. Margarines made from palm, rapeseed, coconut, and sunflower are more frequently used in the formulation of GFPs, especially in the case of bread and similar, biscuits, sweets and semi-sweets, and pastries and cakes. Eighty five percent of gluten-free pastries and cakes include added emulsifiers, which are absent in GCCs. The frequent use of additives of a fatty nature in all the breakfast products studied, including bread, is remarkable, except for gluten-containing and gluten-free churros.

The types of sugars and sweeteners and the frequency of use in the formulation of GFPs and GCCs commonly consumed at breakfast is shown in Table 4. A significant number of all food groups consumed include a wide variety of sugars, particularly sucrose, dextrose, glucose, and fructose syrup. Except for churros and gluten-containing pastries and cakes, between 60 to 85% of sweet breakfast products contain added sucrose, both GFPs as well as GCCs. Sucrose addition to bread, both gluten-free and regular, is also frequent (45% of products). Dextrose is significantly more frequently used in gluten-free bread and similar, pastries and cakes, as compared to GCCs.

Table 5 includes the fibre-type ingredients used in GFPs and GCCs. Fibre is more frequently added to GFPs compared to GCCs, especially in the case of bread and similar and pastries and cakes. These breakfast products mainly include hydroxypropyl methyl cellulose, guar gum, and xanthan gum. In addition, psyllium and bamboo are found in the bread and similar group.

### 3.2. Energy and Nutrient Composition of Gluten-Free and Gluten-Containing Breakfast Products

Table 6 shows the average energy content and nutrient composition of foods typically consumed for breakfast among Spanish children and adolescents, i.e., bread, breakfast cereals, biscuits, bakery products, and churros. Only gluten-free breakfast cereals show no nutritional differences with their GCCs. However, gluten-free breads contain a higher amount of fat and saturated fat, sugars, fibre, and salt, and a lower amount of protein, compared to GCCs. Gluten-free biscuits provide a higher amount of carbohydrates and a lower amount of protein; gluten-free pastries and cakes provide less energy, sugars, and protein, but have increased fibre and salt contents. Finally, churros provide less protein and salt, as compared to GCCs.

### 3.3. Food Habits and Nutrient Intakes for Breakfast

Table 7 shows the daily intake of energy, macronutrients, fibre and salt in Spanish children and adolescents with CD compared to controls, obtained from both the total diet and from breakfast only. The percentage contribution of the energy and nutrient intake for breakfast to total daily intake is also provided. Children and adolescents with CD consumed significantly less energy at breakfast as compared to controls, and the contribution of breakfast to the total daily energy was also slightly lower (19 vs. 20%), although not significantly. Similarly, the intake of saturated fatty acids in breakfast and the contribution of this meal to total saturated fatty acid intakes was smaller in CD. On the other hand, breakfast had a higher contribution to total salt intake in the infant–juvenile celiac population, but daily salt intake was significantly lower compared to controls. As for protein, daily intake and protein contained in breakfast were lower in children and adolescents with CD. We found no differences in carbohydrates, sugars, or fibre intakes.

Table 7 also represents the macronutrient distribution for total daily energy intake and for the energy obtained from breakfast in children and adolescents with CD. Macronutrient contribution to total daily energy intake in both groups was similar, except for protein, which was lower in the case of participants with CD. In the case of breakfast, carbohydrates provided a higher proportion of energy and a lower amount of proteins in participants with CD, as compared to controls.

Table 8 shows the type of products consumed for breakfast by Spanish children and adolescents with CD compared to controls. The four main food groups most frequently consumed at breakfast were grain products, sugars, sweets and pastries, milk and dairy products, and fruits, and there were no significant differences between CD and controls. However, the consumption of eggs and derivatives was more frequent among children and adolescents with CD.

Table 9 shows the contribution of the four main breakfast food groups (grain products, fruits, dairies, sugars, sweets, and confectionery) to the energy and nutrients provided by this meal in both children and adolescents with CD and controls. The other food groups consumed at breakfast (eggs, meat products, vegetables, oils, etc.) contribute only a small proportion of the energy and nutrients and are, therefore, not shown. The percentage of energy and nutrients provided by fruits at breakfast was very similar between participants with CD and controls. Milk and dairy products contributed more to saturated fat and protein intakes at breakfast, and less to the salt intake in the CD group as compared to controls. Major differences were found in the grain products (grains and flours, breakfast cereals, breads, biscuits, and baked goods) and sugars, sweets, and confectionery (sugars, jams, chocolates, sweets, and pastries). The products belonging to the group of sugars, sweets, and confectionery contribute to energy and nutrient intakes in breakfast at a greater extent for the group of children and adolescents with CD and, in contrast, foods from the grain products group contribute more extensively for the control group. Therefore, although the total amount of carbohydrates and simple sugars consumed at breakfast was similar between the two groups, their origin and nature are different.

## 4. Discussion

In this study, we have extensively analysed the breakfast diet among Spanish children and adolescents with celiac disease (CD) in comparison with a control sample (non-celiac). Most of the children and adolescents evaluated in both groups ate breakfast every day (98.5%). This was a positive observation, since breakfast consumption compared to skipping breakfast has been associated with better nutrient intake in different studies [1,4,7,38,39,40,41]. The group of children and adolescents with CD ingested slightly less energy (not significant, 19 vs. 20% of daily energy) than controls. The intake of energy for breakfast in our study is slightly higher, and therefore better, than that reported by Ruiz et al. (2018) in the ANIBES Study on the general Spanish population [2], in which children and teenagers only consumed 18% of daily energy in this meal, and similarly to the data reported by Cuadrado Soto et al. (2020) from the ENALIA Study [1], in which more than half of the children who ate breakfast (56.4%) obtained less than 20% of their daily calories at breakfast, with a mean of 18.3%. According to current Spanish dietary guidelines and the International Breakfast Research Initiative (IBRI) recommendations [18,42], breakfast should provide 15–25% of total energy in the diet (circa 300–500 kcal). Therefore, caloric recommendations for breakfast seem to be accomplished in the assessed population groups in the present study.

In terms of adequacy, the macronutrient distribution for breakfast in CD was balanced (54% of total energy from carbohydrates; 12% from proteins; 34% from lipids), with a higher proportion of energy resulting from carbohydrates and a lower proportion from lipids than that which was obtained for the daily value, and for the control group, but very close to the IBRI recommendations [42]. Therefore, children and adolescents with CD do manage to have a balanced breakfast.

In terms of food variety, various studies indicate that breakfast should include foods from at least three key food groups, namely: starchy foods (cereals, pasta, bread), fruit and vegetables, and milk and dairy products [1,2,42]. In this sense, studies carried out on Spanish children found that breakfast at these ages should be improved. For example, the ALADINO Study [12] showed that breakfasts that included foods from the three recommended groups are scarce (only 2.2% of schoolchildren). In the ENALIA Study [1,2,42], the frequency of consumption of the three types of food is higher (8.4%) but is still insufficient. The present study shows more positive data, since almost half of the population sample of both groups took food from all three basic groups, and 49% of participants with CD and 52% of the control group consumed fruit for breakfast. However, the contribution of fruit to breakfast energy and nutrients is still lower than recommended and is similar between subjects with CD and controls. Milk and its derivatives and cereals were present in 96–100% of breakfasts.

When compared to controls, children and adolescents with CD consume a significantly lower amount of protein, both daily and for breakfast, although it is still enough to cover protein needs. This could be due to the statistically lower protein content of the gluten-free products (GFPs), which is the result of the use of corn and rice flours, and corn starch, which have a low protein and high carbohydrate concentration than wheat flour. These results agree with similar studies that indicated that GFPs, compared to gluten-containing counterparts (GCCs), contain lower protein and higher carbohydrate contents [20,34,43,44,45,46,47,48,49,50,51,52]. Breakfast protein is mainly provided by dairy products, especially in celiacs who consume less from cereals, in which gluten is removed. In this sense, some commercialised GFPs have different protein concentrates or isolates (obtained from microorganisms, animals, and plants) that are added to improve both the quality and the nutritional profile of GFPs [53]. It should be noted that, mainly in gluten-free breads, the functionality of the proteins is more relevant than the nutritional properties since trying to mimic the attributes of gluten from diverse protein origins is a technological challenge and a wide research field [52,54,55,56,57,58,59]. In our study, the CD group also ate significantly more eggs and egg products, which are good sources of protein at breakfast. The addition of eggs at breakfast can contribute to nutrient intakes and overall dietary adequacy and play a role in public health initiatives aimed at increasing the intake of under-consumed nutrients and nutrients of concern [60]. This recommendation could be especially interesting to youngsters with CD since eggs would be very useful in forming the structural doughs through improving the cohesion and elasticity of gluten-free breads when low doses are incorporated, as well as increasing the nutritional value [54,61]. Eggs are also widely used in gluten-free bakery and confectionery products to technologically compensate for the withdrawal of gluten, and occasionally in gluten-free breads. Similarly, dairy protein sources, such as yoghurt and cheese, are also added as confirmed by results from other studies [57,61]. In addition, an adequate protein intake could be an advantage in terms of inducing greater satiety and avoiding possible snacking with unsuitable foods in the mid-morning meal.

GFPs may have a lower protein content, but thanks to them, children and adolescents with CD manage to consume enough carbohydrates and fibre in their breakfast, with even a significantly higher contribution of carbohydrates as compared to controls, although they do have to avoid common cereal-based products. The mean carbohydrate and sugar contents of breakfast GFPs were like those of GCCs, except for slightly higher amounts of total carbohydrate in gluten-free biscuits, sweets and semi-sweets and sugars in bread and similar [44,50,52]. Only a few studies have revealed a lower content in total carbohydrates and sugars with significant differences of gluten-free cakes, muffins, pastries, and biscuits compared to those made with wheat flour [62]. As for fibre, it is a common and widespread ingredient used in the formulation of GFPs [52,54,63], as we have demonstrated in this study. Incorporating foreign fibres, or ingredients with a high fibre content, has significantly improved the nutritional composition of GFPs since the offer of commercial gluten-free whole-grain products is very unusual [56]. Pseudo-cereals such as amaranth, buckwheat, and quinoa, but also milled legumes, seeds, and nuts, are optional ingredients increasingly used for the preparation of gluten-free baked goods compared to GCCs [54,56,64], which enlarge the quantity of fibre of the GFPs. Nevertheless, despite the higher fibre content of GFPs, people following a strict GFD have a lower fibre intake than the rest of the population if the intake of the whole day, not only for breakfast, is assessed [27,36,45,65].

Moreover, the intake of saturated fatty acids (SFA) at breakfast appears to be lower in the celiac group, although there is no difference in the intake of these fatty acids in the total diet compared to the non-celiac group, indicating a possible “intake compensation” with other meals of the day. Furthermore, the contribution of SFA to total daily energy intake and that provided for breakfast is similar between the two groups. In this respect, it is important to point out that the content of SFA in cereal-based GFPs depends on the type of oil/fat used for its preparation [43]. In our study of ingredients, we found that the use of polyunsaturated fats, such as sunflower or olive oils, is frequent in breads and pastries and cakes, but the use of saturated fat, such as palm and cocoa is also frequent in pastries and cakes and biscuits, sweets, and semi-sweets. Results are in accordance with most of the previous studies [44,46,47,52,66]. Therefore, saturated fat intake in breakfast in CD may highly depend on the type of products chosen. In addition, it should be noted that the foods that contribute most to the intake of SFA in the celiac group are milk and dairy derivatives, and this food group appears to have a beneficial effect on cardiometabolic risk factors, compared to other sources of SFA [67].

There are some areas for improvement in the formulation of gluten-free breakfast products and in the general habits and food choices for breakfast in children and adolescents with CD [44,45,54,68]. For example, the consumption of carbohydrates, sugars, and fibre was similar in both types of breakfast and is mainly provided by cereals, but celiacs obtained a higher proportion of the aforementioned nutrients from the group of sugars, sweets, and confectionary, and less from grain products, which changes the type of ingredients/nutrients they consume. Generally, the gluten-free diet is rich in products with a high glycaemic index (GI), which increases the development of chronic diseases. In this context, it is also important that these products have high amounts of protein and fibre to lower the GI. It is recommended that the addition of whole grain flours, or pseudocereals and legume could enhance the nutritional quality of GFPs [69]. The products marketed as GFPs, evaluated in our study, were more frequently added with sugars such as dextrose, the syrup of glucose, and non-refined or cane sugar, rice syrup, etc. The inclusion of these sugars, as a fermentable ingredient in GFPs, compensate for the lack of hydrolytic enzymes in starch-based preparations [52]. In contrast, in GCPs, sugars often result from the activity of amylase enzymes on starch. In addition, sugar in GFPs enhances the aroma due to non-enzymatic browning reactions [70]. It is known that, in GCPs, sugar, and also fat, hinder the gluten network [52]. If a diet should require avoiding grain-based products because of their gluten content, children and adolescents could make healthier choices of carbohydrates and fibre sources that provide less sugars, such as nuts, dried fruit, date, or peanut pastes. Added sugars are key nutrients in product reformulation which should be focussed on [52].

According to the total diet data, celiac children and adolescents consume significantly less salt than controls; however, at breakfast, the salt intake is higher. An analysis of products marketed as gluten-free indicates that most of them have a higher salt content, especially breads, pastries, and cakes, compared to GCCs [44,71,72]. Salt reduction in cereal-based products for celiacs is another important issue to be addressed by the food industry, as we have previously proposed [27].

In our view, the nutritional quality of the gluten-free breakfast could also improve with nutrition education, especially focusing children and teenagers with CD. Actual consumption trends in CD warrant the need to promote the consumption of unprocessed GFPs such as pseudocereals, with better nutritional quality, and homemade products with flours different to rice or corn, together with proper nutritional guidance, including the avoidance of manufactured GFPs. The challenge in using unconventional flours in food preparations is the need for high food literacy (e.g., food skills, budgeting, and nutrition knowledge) and more time for meal planning and cooking in comparison to purchasing ready-to-eat products. It would also be helpful to include foods of a different nature, such as more fruits, other sources of protein, and products derived from legumes and nuts.

### Strengths and Limitations

It is the first time that such a detailed study of breakfast in children and adolescents with CD has been carried out, both in terms of food, ingredients, and nutrients. Only by conducting this analysis can differences in the type and nature of nutrients be observed, which could be used to assess the quality of breakfasts and to establish new indices in the future.

Unfortunately, we have not been able to analyse the micronutrient intake of breakfast in both groups, celiacs and controls, since micronutrient content is not specified on the labelling of GFPs. Because GFPs constitute an important part of the diet of young people with CD (they provide up to a quarter of the daily energy), we would be significantly underestimating the micronutrient intake. Another limitation of the study was the small sample size, because the research is a follow-up analysis of an initial study comparing the dietary habits of children diagnosed with CD, and the sample size was adapted to the present study to compare breakfast habits and nutritional quality. Future studies which include a larger sample size will further contribute to this area of research.

## 5. Conclusions

The present study provides information on the type of breakfast eaten by a sample of Spanish children and adolescents with celiac disease (CD) compared to children of the same age without the disease (controls). Until now, breakfast has not been evaluated in this population group and, according to the literature, this meal is related to a better nutritional status, the prevention of cardiometabolic diseases, and the improvement of cognitive performance. Relevant positive issues were observed, such as that virtually no one skips breakfast, and almost half of the sample includes all three recommended food groups (cereals, dairy, and fruits). The energy intake and the distribution of macronutrients is quite adequate. However, there are also many areas for improvement. For example, the cereal-based gluten-free products that are normally included in breakfasts in our study are almost 100% manufactured by the food industry. Thus, we observed that although commercial gluten-free products (GFPs) contribute to an adequate intake of carbohydrates and fibre, they also provide less protein and more added sugars than GCCs. Moreover, the group with CD has a higher intake of nutrients from the group of “sugars, sweets, and confectionary” than those provided by grain products. To compensate for the low protein intake from this source, celiacs consume more protein from dairy products and seem to include more eggs in this meal of the day compared to controls.

## Figures and Tables

**Table 1 nutrients-15-02368-t001:** Types of flour and frequency of use in the formulation of gluten-free and gluten-containing products commonly consumed for breakfast.

Breakfast Product		*n*	Rice *n* (%)	Corn *n* (%)	Millet *n* (%)	Amaranth *n* (%)	Legumes *n* (%)	Nut *n* (%)	Wheat *n* (%)	Whole Meal Wheat *n* (%)	Rye *n* (%)	Barley *n* (%)	Malt *n* (%)	Oat *n* (%)	Linseed *n* (%)
Bread and similar	GFPs GCCs	100 101	62 (62.0) *** 14 (14.0)	21(21.0) *** 3 (3.0)	16 (16.5) ** 4 (4.0)	1 (2.4) 0 (0.0)	10 (10.5) ** 25 (24.8)	1 (1.0) 0 (0.0)	0 (0.0) *** 89 (88.1)	0 (0.0) 6 (5.9)	1 (1.0) *** 25 (24.8)	0 (0.0) 1 (0.0)	0 (0.0) *** 22 (21.8)	0 (0.0) 4 (4.0)	11 (11.3) 18 (17.8)
Breakfast cereals	GFPs GCCs	35 30	15 (42.9) 10 (33.3)	27 (77.1) * 14 (46.7)	0 (0.0) 0 (0.0)	3 (8.6) 0 (0.0)	1 (2.9) 0 (0.0)	3 (8.6) 0 (0.0)	0 (0.0) *** 12 (40.0)	0 (0.0) *** 11 (36.7)	0 (0.0) 0 (0.0)	0 (0.0) 3 (10.0)	1 (2.9) ** 9 (30.0)	5 (14.3) * 11 (36.7)	1 (2.9) 0 (0.0)
Biscuits, sweets, and semi-sweets	GFPs GCCs	96 95	55 (57.3) *** 8 (8.4)	61(63.5) *** 3 (3.2)	0 (0.0) 0 (0.0)	0 (0.0) 0 (0.0)	24 (25.0) *** 2 (2.1)	0 (0.0) 0 (0.0)	0 (0.0) *** 95 (100.0)	0 (0.0) *** 16 (16.8)	0 (0.0) 0 (0.0)	0 (0.0) 0 (0.0)	0 (0.0) 5 (5.3)	0 (0.0) 0 (0.0)	0 (0.0) 0 (0.0)
Pastries and cakes	GFPs GCCs	127 116	48 (37.8) *** 6 (5.2)	17(13.4) *** 0 (0.0)	0 (0.0) 0 (0.0)	0 (0.0) 0 (0.0)	28 (22.0) *** 5 (4.3)	15(11.8) *** 0 (0.0)	0 (0.0) *** 113 (97.4)	0 (0.0) 0 (0.0)	1 (0.8) 3 (2.6)	0 (0.0) 3 (2.6)	0 (0.0) 0 (0.0)	0 (0.0) ** 9 (7.8)	3 (2.4) 0 (0.0)
Churros	GFPs GCCs	6 6	0 (0.0) 0 (0.0)	0 (0.0) 0 (0.0)	0 (0.0) 0 (0.0)	0 (0.0) 0 (0.0)	0 (0.0) 1 (16.7)	0 (0.0) 0 (0.0)	0 (0.0) *** 6 (100.0)	0 (0.0) 0 (0.0)	0 (0.0) 0 (0.0)	0 (0.0) 0 (0.0)	0 (0.0) 0 (0.0)	0 (0.0) 0 (0.0)	0 (0.0) 0 (0.0)

Results are expressed as frequency (*n*) of products, including a specific ingredient, and percentage based on the total products within the group. ** p* < 0.05 ** *p* < 0.01 *** *p* < 0.001 gluten-free products (GFPs) vs. gluten-containing counterparts (GCCs) within the same food group. Legumes: legumes, carob. Malt: malt, barley malt, rye malt, and maize malt.

**Table 2 nutrients-15-02368-t002:** Types of starch and frequency of use in the formulation of gluten-free and gluten-containing products commonly consumed for breakfast.

Breakfast Product		*n*	Corn *n* (%)	Rice *n* (%)	Potato *n* (%)	Tapioca *n* (%)	Modified *n* (%)	Wheat *n* (%)
Bread and similar	GFPs GCCs	100 101	91 (91.0) *** 11 (10.9)	28 (28.0) *** 0 (0.0)	7 (7.0) * 1 (1.0)	16 (16.0) *** 2 (2.0)	1 (1.0) 1 (1.0)	0 (0.0) * 6 (5.9)
Breakfast cereals	GFPs GCCs	35 30	0 (0.0) 1 (3.3)	0 (0.0) 0 (0.0)	0 (0.0) 0 (0.0)	0 (0.0) 0 (0.0)	0 (0.0) 1 (3.3)	0 (0.0) 1 (3.3)
Biscuits, sweets, and semi-sweets	GFPs GCCs	96 95	59 (61.5) *** 1 (1.1)	26 (27.1) *** 1 (1.1)	29 (30.2) *** 0 (0.0)	4 (4.2) * 0 (0.0)	0 (0.0) 0 (0.0)	0 (0.0) *** 13 (13.7)
Pastries and cakes	GFPs GCCs	127 116	108 (85.0) *** 10 (8.6)	29 (22.8) *** 0 (0.0)	27 (21.3) *** 3 (2.6)	12 (9.4) ** 0 (0.0)	2 (1.6) * 8 (6.9)	1 (0.8) 5 (4.3)
Churros	GFPs GCCs	6 6	3 (50.0) * 0 (0.0)	0 (0.0) 0 (0.0)	0 (0.0) 0 (0.0)	0 (0.0) 0 (0.0)	0 (0.0) 0 (0.0)	0 (0.0) 0 (0.0)

Results are expressed as frequency (*n*) of products, including a specific ingredient, and percentage based on the total products within the group. ** p* < 0.05 ** *p* < 0.01 *** *p* < 0.001 gluten-free products (GFPs) vs. gluten-containing counterparts (GCCs) within the same food group.

**Table 3 nutrients-15-02368-t003:** Types of fat and frequency of use in the formulation of gluten-free and gluten-containing products commonly consumed for breakfast.

Breakfast Product		*n*	Sunflower *n* (%)	Palm *n* (%)	Olive *n* (%)	Cocoa *n* (%)	Rapeseed Oil *n* (%)	Margarine 1 *n* (%)	Margarine 2 *n* (%)	Coconut Oil *n* (%)	Animal Fat *n* (%)	Emulsifiers *n* (%)
Bread and similar	GFPs GCCs	100 101	70 (70.0) ** 49 (48.5)	5 (5.0) 4 (4.0)	13 (13.0) 10 (9.9)	1 (1.0) 1 (1.0)	7 (7.1) 3 (3.0)	1 (1.0) 1 (1.0)	29 (29.0) *** 2 (2.0)	27 (27.0) *** 4 (4.0)	0 (0.0) 1 (1.0)	59 (59.0) 51 (50.5)
Breakfast cereals	GFPs GCCs	35 30	10 (28.6) 10 (33.3)	4 (11.4) 2 (6.7)	0 (0.0) 0 (0.0)	12 (34.3) * 4 (13.3)	1 (2.9) 3 (10.0)	0 (0.0) 0 (0.0)	0 (0.0) 0 (0.0)	1 (2.9) 4 (13.3)	0 (0.0) 0 (0.0)	13 (37.1) 11 (36.7)
Biscuits, sweets, and semi-sweets	GFPs GCCs	96 95	34 (35.4) 39 (41.1)	39 (40.6) 43 (45.3)	11 (11.5) 9 (9.5)	41 (42.7) 42 (44.2)	4 (4.2) 1 (1.1)	13 (13.5) ** 1 (1.1)	5 (5.2) 1 (1.1)	15 (15.6) 9 (9.5)	48 (50.0) 38 (40.0)	38 (39.6) * 52 (54.7)
Pastries and cakes	GFPs GCCs	127 116	95 (74.8) 88 (75.9)	35 (27.6) * 47 (40.9)	8 (6.3) 3 (2.6)	61 (48.4) 67 (57.8)	7 (5.5) 9 (7.8)	28 (22.0) *** 0 (0.0)	17 (13.4) 12 (10.3)	17 (13.4) 14 (12.1)	19 (15.0) *** 0 (0.0)	107 (84.9) *** 0 (0.0)
Churros	GFPs GCCs	6 6	2 (33.3) 2 (33.3)	1 (16.7) 0 (0.0)	0 (0.0) 0 (0.0)	0 (0.0) 0 (0.0)	1 (16.7) 0 (0.0)	0 (0.0) 0 (0.0)	1 (16.7) 1 (16.7)	0 (0.0) 0 (0.0)	0 (0.0) 0 (0.0)	0 (0.0) 2 (33.3)

Results are expressed as frequency (*n*) of products, including a specific ingredient, and percentage based on the total products within the group. ** p* < 0.05 ** *p* < 0.01 *** *p* < 0.001 gluten-free products (GFPs) vs. gluten-containing counterparts (GCCs) within the same food group. Cocoa: cocoa oil, cocoa butter, cocoa, cocoa paste, and chocolate powder. Margarine 1: palm, rapeseed, and emulsifier. Margarine 2: coconut and sunflower. Animal fat: animal fat, butter, and milk fat. Emulsifiers: mono- and diglycerides of fatty acids and sunflower, soy, and rapeseed lecithin.

**Table 4 nutrients-15-02368-t004:** Types of sugars and sweeteners and frequency of use in the formulation of gluten-free and gluten-containing products commonly consumed for breakfast.

Breakfast Product		*n*	Sucrose *n* (%)	Dextrose *n* (%)	Glucose and Fructose Syrup *n* (%)	Non-Refined or Cane Sugar *n* (%)	Rice Syrup *n* (%)	Beetroot Sugar Syrup *n* (%)	Honey *n* (%)	Lactose *n* (%)	Other Sugars *n* (%)	Low Calorie Sweetener *n (%)*
Bread and similar	GFPs GCCs	100 101	45 (45.0) 45 (44.6)	32 (32.0) *** 4 (4.0)	11 (11.0) 4 (4.0)	8 (8.0) * 2 (2.0)	22 (22.0) *** 1 (1.0)	1 (1.0) 0 (0.0)	5 (5.0) * 0 (0.0)	1 (1.0) 0 (0.0)	51 (51.0) *** 10 (9.9)	0 (0.0) 0 (0.0)
Breakfast cereals	GFPs GCCs	35 30	23 (65.7) 25 (83.3)	2 (5.7) 1 (3.3)	4 (11.4) ** 12 (40.0)	6 (17.1) 5 (16.7)	1 (2.9) 0 (0.0)	0 (0.0) 0 (0.0)	1 (2.9) * 6 (20.0)	1 (2.9) 0 (0.0)	8 (22.9) *** 21 (70.0)	1 (2.9) 0 (0.0)
Biscuits, sweets, and semi-sweets	GFPs GCCs	96 95	82 (85.4) 81 (85.3)	15 (15.6) 12 (12.6)	32 (33.3) *** 0 (0.0)	20(31.7) *** 6 (6.3)	1 (1.0) 0 (0.0)	9 (9.4) ** 0 (0.0)	2 (2.1) 2 (2.1)	29(30.2) ** 48 (50.5)	46 (59.7) 57 (60.0)	4 (4.2) 6 (6.3)
Pastries and cakes	GFPs GCCs	127 116	114 (89.8) 108 (93.1)	49 (38.6) ** 27 (23.3)	61 (48.0) 63 (54.3)	6 (4.7) 2 (1.7)	3 (2.4) 0 (0.0)	0 (0.0) 0 (0.0)	1 (0.8) 3 (2.6)	6 (4.7) * 0 (0.0)	95 (74.8) * 73 (62.9)	15 (11.8) * 25 (21.6)
Churros	GFPs GCCs	6 6	0 (0.0) 1 (16.7)	2 (33.3) 2 (33.3)	0 (0.0) 0 (0.0)	0 (0.0) 0 (0.0)	0 (0.0) 0 (0.0)	0 (0.0) 0 (0.0)	0 (0.0) 0 (0.0)	0 (0.0) 0 (0.0)	2 (33.3) 3 (50.0)	0 (0.0) 0 (0.0)

Results are expressed as frequency (*n*) of products, including a specific ingredient and percentage based on the total products within the group. * *p* < 0.05 ** *p* < 0.01 *** *p* < 0.001 gluten-free products (GFPs) vs. gluten-containing counterparts (GCCs) within the same food group. Non-refined or cane sugar: molasses, cane sugar, and cane sugar syrup. Lactose: lactose and milk powder. Other sugars: isomaltose, fructose, glucose, agave syrup, corn syrup, barley malt extract, caramelised sugar syrup, invert sugar syrup, and liquid caramel.

**Table 5 nutrients-15-02368-t005:** Types of fibres and frequency of use in the formulation of gluten-free and gluten-containing products commonly consumed for breakfast.

Breakfast Product		*n*	Hydroxypropyl Methyl Cellulose *n* (%)	Xanthan Gum *n* (%)	Guar Gum *n* (%)	Gum *n* (%)	Psyllium Bamboo *n* (%)	Sodium Carboxymethyl Cellulose *n* (%)	Pectin *n* (%)	Other Fibres *n* (%)	Oat Fibre Wheat Bran *n* (%)
Bread and similar	GFPs GCCs	100 101	67 (67.0) *** 5 (5.0)	48 (48.0) *** 6 (5.9)	14 (14.0) 17 (16.8)	36 (36.0) * 23 (22.8)	48 (48.0) *** 3 (3.0)	14 (14.0) ** 3 (3.0)	11 (12.4) 1 (7.7)	19 (19.8) *** 4 (4.0)	0 (0.0) *** 13 (12.9)
Breakfast cereals	GFPs GCCs	35 30	0 (0.0) 0 (0.0)	0 (0.0) 0 (0.0)	0 (0.0) 0 (0.0)	0 (0.0) 0 (0.0)	1 (2.9) 0 (0.0)	0 (0.0) 0 (0.0)	0 (0.0) 0 (0.0)	1 (3.1) 0 (0.0)	0 (0.0) * 5 (16.7)
Biscuits, sweets, and semi-sweets	GFPs GCCs	96 95	0 (0.0) 0 (0.0)	23 (24.0) *** 0 (0.0)	20 (20.8) *** 1 (1.1)	4 (4.2) 2 (2.1)	0 (0.0) 0 (0.0)	0 (0.0) 0 (0.0)	11 (18.0) 8 (8.4)	0 (0.0) 0 (0.0)	0 (0.0) 0 (0.0)
Pastries and cakes	GFPs GCCs	127 116	31 (24.4) *** 0 (0.0)	91 (71.7) *** 22 (19.0)	26 (20.5) * 13 (11.2)	98 (77.2) *** 0 (0.0)	16 (12.6) *** 0 (0.0)	13 (10.2) ** 2 (1.7)	9 (7.1) ** 0 (0.0)	4 (4.0) * 0 (0.0)	0 (0.0) 0 (0.0)
Churros	GFPs GCCs	6 6	0 (0.0) 0 (0.0)	2 (33.3) 1 (16.7)	1 (16.7) 0 (0.0)	0 (0.0) 1 (16.7)	0 (0.0) 0 (0.0)	0 (0.0) 0 (0.0)	0 (0.0) 0 (0.0)	0 (0.0) 0 (0.0)	0 (0.0) 0 (0.0)

Results are expressed as frequency (*n*) of products, including a specific ingredient, and percentage based on the total products within the group. * *p* < 0.05 ** *p* < 0.01 *** *p* < 0.001 gluten-free products (GFPs) vs. gluten-containing counterparts (GCCs) within the same food group. Gum: unspecified gum. Pectin: pectin, fibre of apple, banana, and citrus. Other fibres: chicory, potato, rice, pea, soy fibre and rice, and pea bran.

**Table 6 nutrients-15-02368-t006:** Energy and nutrient composition per 100 g of gluten-free and gluten-containing breakfast products, according to labelling nutritional information.

Breakfast Products		*n*	Energy (kcal)	Fats (g)	Saturated Fat (g)	Carbohydrates (g)	Sugars (g)	Protein (g)	Fibre (g)	Salt (g)
Bread and similar	GFPs GCCs	100 101	294.9 ± 57.8 288.4 ± 52.1	5.6 ± 3.4 * 4.0 ± 3.4	2.2 ± 2.6 * 1.1 ± 1.4	55.6 ± 14.2 52.8 ± 12.1	5.1 ± 3.3 * 3.9 ± 2.9	3.0 ± 1.9 * 9.2 ± 2.8	5.6 ± 2.1 * 3.9 ± 2.2	1.4 ± 0.5 * 1.3 ± 0.4
Breakfast cereals	GFPs GCCs	35 30	385.1 ± 26.5 388.5 ± 29.9	4.5 ± 4.5 6.0 ± 5.2	1.3 ± 1.4 2.0 ± 2.4	75.6 ± 9.3 71.4 ± 10.1	15.4 ± 10.8 19.4 ± 9.5	7.9 ± 2.6 8.7 ± 2.3	5.3 ± 3.4 6.9 ± 4.7	0.6 ± 0.6 0.6 ± 0.4
Biscuits, sweets, and semi-sweets	GFPs GCCs	96 95	471.5 ± 41.8 469.3 ± 48.7	19.9 ± 6.0 20.0 ± 5.8	9.4 ± 5.5 8.7 ± 5.9	67.5 ± 6.4 * 65.1 ± 7.4	25.5 ± 8.4 26.9 ± 10.6	4.4 ± 1.5 * 6.3 ± 1.4	3.9 ± 4.8 3.5 ± 2.0	0.6 ± 0.5 0.7 ± 0.4
Pastries and cakes	GFPs GCCs	127 116	400.7 ± 71.4 * 427.0 ± 72.9	21.5 ± 6.9 22.5 ± 6.7	7.3 ± 5.1 8.7 ± 6.1	47.0 ± 8.2 52.1 ± 34.9	20.5 ± 8.8 * 23.7 ± 10.6	4.1 ± 1.8 * 5.6 ±1.5	3.0 ± 1.7 * 2.3 ± 1.4	0.8 ± 0.6 * 0.6 ± 0.3
Churros	GFPs GCCs	6 6	237.9 ± 122.2 201.5 ± 115.6	9.3 ± 8.7 4.1 ± 8.3	3.5 ± 4.2 1.8 ± 4.0	36.6 ± 11.1 35.0 ± 10.4	6.5 ± 10.9 8.6 ± 10.3	1.7 ± 1.6 * 5.2 ± 0.9	- 2.0 ± 0.3	0.8 ± 0.2 * 17.3 ± 39.5

Data are expressed as average ± standard deviation. * *p* < 0.05 gluten-free products (GFPs) vs. gluten-containing counterparts (GCCs) within the same food group.

**Table 7 nutrients-15-02368-t007:** Contribution of gluten-free and gluten-containing breakfast products to the diet (energy and nutrient content) and macronutrient distribution for total daily energy intake and energy from breakfast in Spanish children and adolescents with celiac disease.

	Total Daily Intake		Intake from Breakfast		% Contribution of Breakfast	
	CD	CONTROL	CD	CONTROL	CD	CONTROL
	*n* = 70	*n* = 67	*n* = 70	*n* = 67	*n* = 70	*n* = 67
Energy (kcal/day)	2043.0 ± 449.1	2121.4 ± 469.4	370.0 ± 107.2 *	411.7 ± 115.8	18.8 ± 6.5	20.1 ± 6.7
Fats (g/day)	93.9 ± 20.1	99.1 ± 26.8	14.0 ± 6.3 *	16.2 ± 7.0	15.5 ± 7.6	17.4 ± 8.6
% Energy from fats	41.9 ± 6.4	42.4 ± 7.8	34.1 ± 9.0	34.5 ± 9.0		
Saturated fat (g/day)	32.0 ± 7.9	33.8 ± 10.0	6.3 ± 3.1 *	7.8 ± 3.5	20.2 ± 9.7 *	24.9 ± 12.4
% Energy from saturated fat	14.3 ± 2.5	14.1 ± 3.3	15.1 ± 5.4	16.7 ± 5.3		
Carbohydrates (g/day)	208.8 ± 66.6	216.5 ± 57.6	49.5 ± 15.3	52.1 ± 15.2	25.4 ± 10.0	24.9 ± 7.5
% Energy from carbohydrates	40.6 ± 7.0	40.9 ± 6.6	53.9 ± 7.7 *	51.1 ± 7.3		
Sugars (g/day)	88.5 ± 25.3	88.5 ± 26.6	27.1 ± 10.5	30.4 ± 11.2	32.2 ± 13.5	35.9 ± 13.5
Protein (g/day)	77.4 ± 18.1 *	89.0 ± 20.5	10.30 ± 2.6 *	13.0 ± 4.0	14.1 ± 5.2	15.1 ± 5.0
% Energy from protein	15.2 ± 2.2 *	16.8 ± 2.5	11.6 ± 3.7 *	13.1 ± 3.0		
Fibre (g/day)	18.0 ± 7.6	16.9 ± 5.5	2.6 ± 1.5	2.5 ± 1.1	15.9 ± 9.3	15.6 ± 8.1
Salt (g/day)	4.5 ± 2.4 *	5.4 ± 2.2	0.9 ± 0.4	0.8 ± 0.5	23.5 ± 12.6 *	16.2 ± 8.2

Data are expressed as average ± standard deviation. * *p* < 0.05 children and adolescents with celiac disease (CD) vs. control.

**Table 8 nutrients-15-02368-t008:** Food groups consumed for breakfast by Spanish children and adolescents with celiac disease.

Food Groups	CD *n* = 70	CONTROL *n* = 67
Grains (*n* (%))	67 (95.7)	67 (100.0)
Sugars, sweets, and pastries (*n* (%))	59 (84.2)	48 (71.6)
Milk and dairy products (*n* (%))	69 (98.6)	67 (100.0)
Fruits (*n* (%))	34 (48.6)	35 (52.2)
Legumes (*n* (%))	2 (2.9)	2 (3.0)
Vegetables (*n* (%))	6 (8.6)	4 (6.0)
Meat and meat products (*n* (%))	7 (10.0)	14 (20.9)
Fish and derivatives (*n* (%))	1 (1.4)	1 (1.5)
Eggs and derivatives (*n* (%))	7 (10.0)	1 (1.5) *
Oils and fats (*n* (%))	36 (51.4)	28 (41.8)
Beverages (*n* (%))	10 (14.0)	4 (6.0)
Readily prepared and precooked meals (*n* (%))	0	1 (1.5)
Sauces and condiments (*n* (%))	4 (5.7)	0 *

Results are expressed as frequency (*n*) number of subjects taking the product and percentage based on the total number of participants. * *p* < 0.05 children and adolescents with celiac disease (CD) vs. control.

**Table 9 nutrients-15-02368-t009:** Contribution of the different food groups to the energy and nutrient content of breakfast in Spanish children and adolescents with celiac disease.

	Intake in Breakfast		% Contribution from Grain Products		% Contribution from Fruits		% Contribution from Dairy		% Contribution from Sugars, Sweets, and Confectionery	
	CD *n* = 70	CONTROL *n* = 67	CD *n* = 70	CONTROL *n* = 67	CD *n* = 70	CONTROL *n* = 67	CD *n* = 70	CONTROL *n* = 67	CD *n* = 70	CONTROL *n* = 67
Energy (kcal/day)	370.0 ± 107.2 *	411.7 ± 115.8	37.0 ± 16.6 *	44.7 ± 17.8	6.4 ± 8.2	6.9 ± 8.7	32.5 ± 13.5	30.6 ± 11.0	16.6 ± 15.6 **	8.7 ± 11.9
Fats (g/day)	14.0 ± 6.3	16.2 ± 7.0	27.3 ± 23.7 *	38.0 ± 25.2	0.9 ± 1.8	1.3 ± 3.9	41.9 ± 25.7	35.1 ± 19.3	13.2 ± 17.7 *	6.2 ± 14.4
Saturated fat (g/day)	6.3 ± 3.1 *	7.8 ± 3.5	19.3 ± 20.1 ***	34.0 ± 23.6	0.2 ± 0.5	0.3 ± 0.8	54.4 ± 27.2 *	44.8 ± 21.7	11.5 ± 15.0 **	5.5 ± 11.4
Carbohydrates (g/day)	49.5 ± 15.3	52.1 ± 15.2	46.3 ± 19.2 *	54.3 ± 18.7	9.9 ± 12.6	11.1 ± 13.7	21.0 ± 9.5	21.6 ± 8.9	21.3 ± 17.8 **	11.8 ± 13.8
Sugars (g/day)	27.1 ± 10.5	30.4 ± 11.2	17.5 ± 12.0 ***	25.5 ± 17.6	16.9 ± 21.1	17.7 ± 21.2	39.5 ± 18.2	38.7 ± 16.7	23.7 ± 18.4 **	16.2 ± 15.5
Protein (g/day)	10.30 ± 2.6 *	13.0 ± 4.0	18.0 ± 10.4 ***	28.1 ± 10.6	3.1± 4.2	3.0 ± 4.4	66.1 ± 15.2 ***	58.8 ± 14.9	7.2 ± 7.6 *	4.4 ± 8.0
Fibre (g/day)	2.6 ± 1.5	2.5 ± 1.1	62.5 ± 27.4 *	74.2 ± 24.7	13.6 ± 18.9	16.6 ± 22.6	0.0 ± 0.0	0.0 ± 0.0	20.7 ± 23.7 ***	8.8 ± 10.4
Salt (g/day)	0.9 ± 0.4	0.8 ± 0.5	41.6 ± 20.9	45.4 ± 16.1	0.3 ± 0.7	0.3 ± 0.4	29.9 ± 14.5 ***	39.9 ± 14.8	18.5 ± 18.5 ***	8.3 ± 8.7

Data are expressed as average ± standard deviation. * *p* < 0.05 ** *p* < 0.01 *** *p* < 0.001 children and adolescents with celiac disease (CD) vs. control.

## Data Availability

The data on gluten-free food composition used in this study are openly available in the institutional repository at Universidad San Pablo-CEU (CEU Repositorio Institucional) at http://hdl.handle.net/10637/13562 (Last accessed on 15 November 2022). The data on dietary intakes presented in this study are available upon request from the corresponding author.

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
