# Peer review of "Nutritional Composition of Breakfast in Children and Adolescents with and without Celiac Disease in Spain—Role of Gluten-Free Commercial Products"

_nutrients, 2023, doi:10.3390/nu15102368_

Round 1

Reviewer 1 Report

The manuscript aimed “to assess the composition of processed gluten-free products (GFP) used in breakfast and the overall nutritional quality of breakfast in a group of Spanish children and adolescents with CD”. The manuscript presents a pertinent theme but it it must be improved.

-        Insert the line numbers to facilitate the revision.

ABSTRACT

-        In the abstract “Food intakes were estimated using three 24-h dietary records” – How many days of 24hR did you evaluate?

-        Most participants (98.5%) ate breakfast daily” – control or CD group? As you are comparing, data from both groups should be mentioned.

-        The methods’ description in the abstract should be more accurate.

-        Insert a conclusion in the abstract.

INTRODUCTION

-        2nd paragraph: standardize the references as recommended for MDPI journals.

-        At the end of the 2nd paragraph, you could mention that these challenges can stimulate skipping breakfast. Therefore, there will be a connection between the 2nd and 3rd paragraphs.

-        “The consumption of dairy and plant-based foods could play an important role in the prevention of diabetes (type I and II) and cardiometabolic disorders” – if you are mentioning it about breakfast, clear it. If not, remove this sentence.

-        “Cognitive function could also be affected by eating an adequate breakfast [17]. Its regular consumption was similarly correlated with better academic performance scores [18,19].” – It should not be a paragraph. Join it to another paragraph.

-        Because of cultural dietary habits and food recommendations for breakfast, breakfast is the meal in the day in which gluten-free processed foods are more probably introduced” – In Spain?

-        Mention in the introduction section the prevalence of CD in children/adolescents in Spain and/or worldwide. Also include the potential effects of gluten consumption for this population (mainly in growth and development).

-        “Therefore, the present study aimed to assess the composition of processed gluten-free products (GFP) used in breakfast and the overall nutritional quality of breakfast in a group of Spanish children and adolescents with CD, as compared to a group of similar age and gender characteristics without the disease (control)” – it seems that you evaluate only the composition of processed GFP. It is also important to mention what you consider “processed food” since we have divergencies between the concept used by researchers from food technology and public health.

-        “The ingredient and nutrient composition of GFP, currently available on the market, was evaluated in contrast to their gluten-containing counterparts (GCC).” – It should be placed in the method section.

METHODS

-        not suffering related disorders” – what kind of related disorders?

-        Insert the selection (inclusion and exclusion) criteria of the control group.

-         “The gluten-free products (GFP) composition database available at the Universidad San Pablo-CEU institutional repository [28,29] was used for GFP composition data. This food database was compiled using the nutritional composition and ingredient list data from labels, as previously described [29].” If you use a composition database, do not mention that you used labels. Correct it in all places that you mention that you used labels, because it seems you went to the market to evaluate the labels. Also correct the information “GCC were also collected from the labeling on their packaging”, removing “also”.

-        Ingredients were chosen according to their impact on the nutritional profile of GFP and GCC (starchy ingredients, fats, sugars, and fibre). In particular, the top ten most frequently used ingredients were considered” – Why did you not evaluate all ingredients? Explain better the cutoff point to “choose” the ingredients.

-        Salt and additives are also critical ingredients and are much used in gluten-free products.

-        If the participant mentioned the consumption of a non-industrialized product (prepared or not at home) or starchy products that do not contain gluten (for example, popcorn or other corn products), how did you evaluate it?

-        “Further details of the study protocol have been previously published” – I consider it essential to describe all the methods used in this manuscript. The reader needs to understand how the study was performed.

RESULTS

-        This section would be benefited from table insertion near its mention in the text. Also, a heat map or word clouds could be used to show the ingredients' frequencies. There are some studies previously published that evaluate gluten-free bread, cake, cookies etc. that should be mentioned.

-        “Most participants (98.5%) took breakfast every day, and only one person in each group skipped breakfast once.” – evaluated in the 24H recall? If the participant skips breakfast, shouldn’t he/she be excluded from the sample? Insert the number for celiac and non-celiac groups separately, allowing comparison.

-        “Cereal-based foods marketed for CD patients are products widely consumed by the group of children and adolescents, as recently demonstrated by our previous research [23], and breakfast is one of the meals in which they are most likely incorporated, due to socio-cultural background” – discussion, not result.

-        “Therefore, the present study aimed, on a first basis, to assess ingredient and nutritional differences between gluten-free products (GFP) and gluten-containing counterparts (GCC).” – it is the objective of the study, not a result.

-        Detail the participants’ mean age and gender (%) in the results or in the method section.

-        due to socio-cultural background” – explain it

Thank you for the opportunity to review this manuscript!

none.

Author Response

COMMENTS OF REVIEWER 1
 The manuscript aimed “to assess the composition of processed gluten-free products (GFP) used in breakfast and the overall nutritional quality of breakfast in a group of Spanish children and adolescents with CD”. The manuscript presents a pertinent theme, but it must be improved.
Thank you for reviewing our manuscript and for the assessment. The comments and suggestions have helped us to improve the quality of the work. Please find below a list of changes and detailed answers to every point raised in the review.
We appreciate the opinion of the reviewer on the suitability of the present manuscript to be published in Nutrients.
We would like to comment that we uploaded a last version of the manuscript with activated "Tracked Changes", but the reference lines mentioned to the comments of this review corresponding to the uploaded version “no Tracked Changes”.

DETAILED COMMENTS

ABSTRACT
-       In the abstract “Food intakes were estimated using three 24-h dietary records” – How many days of 24hR did you evaluate?
-    “Most participants (98.5%) ate breakfast daily” – control or CD group? As you are comparing, data from both groups should be mentioned.
-        The methods’ description in the abstract should be more accurate.
-        Insert a conclusion in the abstract.
According to the reviewer’s recommendations, we have carefully revised the abstract highlighting every aspect mentioned above, but considering also that there is a limitation according to the online Nutrients Guide for Authors. Some explanations in the abstract are extendedly detailed in the manuscript to follow. Please, refer to Page 1, lines 22-24 and lines 32-33).
We conducted three 24-h dietary records, i.e., three days.
INTRODUCTION
-        2nd paragraph: standardize the references as recommended for MDPI journals.
According with the reviewer’s puntualisation, we have carefully revised all the references quoted in the text and listed references in the revised manuscript following the clues described in the online Nutrients Guide for Authors.
-        At the end of the 2nd paragraph, you could mention that these challenges can stimulate skipping breakfast. Therefore, there will be a connection between the 2nd and 3rd paragraphs.
Following the reviewer’s suggestion, we connected the two paragraphs (Page 2, lines 60-61).
-        “The consumption of dairy and plant-based foods could play an important role in the prevention of diabetes (type I and II) and cardiometabolic disorders” – if you are mentioning it about breakfast, clear it. If not, remove this sentence.
It is not clearly related to breakfast, but to diet overall, so we have removed the sentence.
-        “Cognitive function could also be affected by eating an adequate breakfast [17]. Its regular consumption was similarly correlated with better academic performance scores [18,19].” – It should not be a paragraph. Join it to another paragraph.
According to the reviewer suggestion, we joined the two paragraphs (Page 2, lines 76-78).
-        “Because of cultural dietary habits and food recommendations for breakfast, breakfast is the meal in the day in which gluten-free processed foods are more probably introduced” – In Spain?
As kindly suggested, we have added that this circumstance occurs in the Spanish population (Page 3, line108).
-       Mention in the introduction section the prevalence of CD in children/adolescents in Spain and/or worldwide. Also include the potential effects of gluten consumption for this population (mainly in growth and development).
Following the reviewer’s suggestion, we have added a paragraph explaining all these issues (Pages 2-3, lines 88-101).
-    “Therefore, the present study aimed to assess the composition of processed gluten-free products (GFP) used in breakfast and the overall nutritional quality of breakfast in a group of Spanish children and adolescents with CD, as compared to a group of similar age and gender characteristics without the disease (control)” – it seems that you evaluate only the composition of processed GFP. It is also important to mention what you consider “processed food” since we have divergencies between the concept used by researchers from food technology and public health.
Thank you for the recommendation, since the text, as it was previously written, could lead to confusion. We have rewritten the paragraph to make it clearer that the analysis of nutrient and ingredient composition of gluten-free products available in the market and their gluten-containing counterparts has only been carried out for processed cereal-based products. However, the nutritional assessment of breakfast is carried on all foods included in the breakfast, both processed and not processed.
On the other hand, we add the reference we have used for the definition of processed products:
Albuquerque TG, Bragotto APA, Costa HS. Processed Food: Nutrition, Safety, and Public Health. Int J Environ Res Public Health. 2022 Dec 7;19(24):16410. doi: 10.3390/ijerph192416410. PMID: 36554295; PMCID: PMC9778909.
Please, refer to Page 3, lines 111-117).
-     “The ingredient and nutrient composition of GFP, currently available on the market, was evaluated in contrast to their gluten-containing counterparts (GCC).” – It should be placed in the method section.
According with the reviewer’s observation, we have placed this sentence in Participants and Methods Section (Page 4, lines 150-151).
METHODS
-        “not suffering related disorders” – what kind of related disorders?
We have modified in the revised manuscript this aspect as follows “…. and not being affected by digestive discomfort at the time of dietary assessment” (Page 2, line 131).
-        Insert the selection (inclusion and exclusion) criteria of the control group.
According to the reviewer’s annotation, we included the selection criteria of the control group (Page 3, lines 134-136).
-         “The gluten-free products (GFP) composition database available at the Universidad San Pablo-CEU institutional repository [28,29] was used for GFP composition data. This food database was compiled using the nutritional composition and ingredient list data from labels, as previously described [29].” If you use a composition database, do not mention that you used labels. Correct it in all places that you mention that you used labels, because it seems you went to the market to evaluate the labels. Also correct the information “GCC were also collected from the labeling on their packaging”, removing “also”.
Considering reviewer´s annotation, we would like to clarify that the food composition database used in the present work was previously developed by our research group (Fajardo et al., 2020), as it is included in the Participants and Methods Section. Therefore, we would also like to explain that nutritional composition and ingredient list data of gluten-free and gluten-containing counterparts’ products was compiled from package labels from major commercial and distribution brand products available in retail stores and supermarkets with the highest market shares in Spain.
  • Fajardo, V.; González, M.P.; Martínez, M.; Samaniego-Vaesken, M.d.L.; Achón, M.; Úbeda, N.; Alonso-Aperte, E. Updated Food composition database for cereal-based gluten free products in Spain: Is reformulation moving on? Nutrients 2020, 12, 2369, https://doi.org/10.3390/nu12082369
-        “Ingredients were chosen according to their impact on the nutritional profile of GFP and GCC (starchy ingredients, fats, sugars, and fibre). In particular, the top ten most frequently used ingredients were considered” – Why did you not evaluate all ingredients? Explain better the cutoff point to “choose” the ingredients.
-        Salt and additives are also critical ingredients and are much used in gluten-free products.
We would like to thank the reviewer for this question. Nutrients considered critical for the health of the population groups who consume them, such as sugars and the type of fat used in their manufacture, were selected. Furthermore, according to previous publications, the products marketed for celiacs have different amounts of the nutrients mentioned, as well as fibre and the flours and starches used when compared with their counterparts; therefore, we also wanted to check whether this was particularly relevant in the foods/products that were most consumed for breakfast by children and young people in Spain.
On the other hand, these were the ingredients and their types that we selected to build the database of gluten-free products that we published in a previous paper (Fajardo et al., 2019), which made it easier for us to compare with the counterparts that are available in supermarkets in our country.
Salt provided by gluten-free products and their comparison with their counterparts for the non-celiac population is indeed included in table 6 (energy and nutrients). The inclusion of additives could be very interesting, but out of the scope of the paper, because of the huge number of different additives employed.
-        If the participant mentioned the consumption of a non-industrialized product (prepared or not at home) or starchy products that do not contain gluten (for example, popcorn or other corn products), how did you evaluate it?
Yes. All products consumed for breakfast were evaluated. The nutritional assessment of the breakfasts of children and adolescents was carried out, as explained in the Participants and Methods Section, with the DIAL® software. If the product consumed by any of the volunteers in the study was included in the database of the software, this information was used, and if it was not included (as was the case of gluten-free products or gluten-free homemade products), it was included in the database using the labelling of the products or the ingredients used to prepare a homemade food (i.e., a sponge cake) according to a recipe book of traditional Spanish gastronomy.
-        “Further details of the study protocol have been previously published” – I consider it essential to describe all the methods used in this manuscript. The reader needs to understand how the study was performed.
Following the reviewer's recommendation, we have added the following sentence: “GFP brands were registered and the composition of all GFP consumed was included in the database of the software used for analysis. As we explained previously [35], labels do not record data on micronutrient composition, therefore, data on micronutrient intake from these products was not quantified.” (Page 4, lines 166-169). The rest of the protocol investigation is not relevant to this article, since it included anthropometric and bone density measures, blood biochemical parameters and physical activity assessment.

RESULTS
-        This section would be benefited from table insertion near its mention in the text. Also, a heat map or word clouds could be used to show the ingredients' frequencies. There are some studies previously published that evaluate gluten-free bread, cake, cookies etc. that should be mentioned.
According with the reviewer’s observation, we inserted each table near its mention in the text.
Besides, we would like to thank your suggestion related to other interesting ways to show the ingredients´ frequencies. However, we think that the tables included in our manuscript show the significance of the specific/numeric data, permitting to know exactly how many times an ingredient is employed in the formulation of the target cereal-based gluten-free products. A heat map or a word cloud do not permit this aspect.
According to your suggestion that some studies previously published evaluating gluten-free bread, cake, cookies, etc. should be mentioned, we would like to emphasize that in the Results Section we consider particularly interesting focusing specifically on our own results, more than in others’.
Besides, and in line with your comment, we had actually mentioned in the Discussion Section some previously published studies evaluating gluten-free bread, cake, cookies, etc. (references nº 19, 33, 42).
  • Mármol-Soler, C.; Matias, S.; Miranda, J.; Larretxi, I.; Fernández-Gil, M.D.P.; Bustamante, M.; Churruca, I.; Martínez, O.; Simón, E. Gluten-free products: do we need to update our knowledge? Foods 2022, 11, 3839, https://doi.org/10.3390/foods11233839
  • Fajardo, V.; González, M.P.; Martínez, M.; Samaniego-Vaesken, M.d.L.; Achón, M.; Úbeda, N.; Alonso-Aperte, E. Updated Food composition database for cereal-based gluten free products in Spain: Is reformulation moving on? Nutrients 2020, 12, 2369, https://doi.org/10.3390/nu12082369
  • Calvo-Lerma, J.; Crespo-Escobar, P.; Martinez-Barona, S.; Fornes-Ferrer, V.; Donat, E.; Ribes-Koninckx, C. Differences in the macronutrient and dietary fibre profile of gluten-free products as compared to their gluten-containing counterparts. J. Clin. Nutr. 2019, 73, 930-936, https://doi.org/10.1038/s41430-018-0385-6

-        “Most participants (98.5%) took breakfast every day, and only one person in each group skipped breakfast once.” – evaluated in the 24H recall? If the participant skips breakfast, shouldn’t he/she be excluded from the sample? Insert the number for celiac and non-celiac groups separately, allowing comparison.
We would like to clarify that following the recommendations by the European Food Safety Authority, diet was estimated using three days 24-h dietary records, as included in the Participants and Methods Section.
The only participant who skips breakfast from each group (CD or control) was not excluded from the study since he/she only skipped breakfast one day from the three days analyzed.
According to reviewer´s recommendation, please refer to Page 4, lines 183-184.
-        “Cereal-based foods marketed for CD patients are products widely consumed by the group of children and adolescents, as recently demonstrated by our previous research [23], and breakfast is one of the meals in which they are most likely incorporated, due to socio-cultural background” – discussion, not result.
-        “Therefore, the present study aimed, on a first basis, to assess ingredient and nutritional differences between gluten-free products (GFP) and gluten-containing counterparts (GCC).” – it is the objective of the study, not a result.
We want to thank you for your comment, and we have removed these sentences from the Results Section.
-        Detail the participants’ mean age and gender (%) in the results or in the method section.
As kindly suggested, we included these data in the Results Section (Page 4, lines 181-183).
-        “due to socio-cultural background” – explain it
We removed this sentence in the revised manuscript.

Reviewer 2 Report

Aim of this paper was to assessed ingredients and nutrient composition of gluten-free breakfast products  and gluten-containing counterparts  and we in a group of Spanish children and adolescents with celiac disease compared to controls.

The paper addresses an interesting issue. In clinical practice it is, in fact, common to have difficulties in the nutritional assessment of the diet in coeliac patients. Especially because comparing breakfast products with gluten and without shows that the latter are often richer in refined flours, sodium, and sugars.

There are some aspects to be improved:

1. there is plagiarism mainly in the methods part (please check the attached file)

2. Reading the paper is difficult. Too many paragraphs, often unconnected. 

3. The number of tables is too many. it would be useful to merge them to make it easier to compare the two groups.  (For example, merge table 7 with table 8)

4. In the tables, insert the p-value where possible. 

5. Such a high figure of adolescents eating breakfast (98%) is astonishing. The figures are much lower. I think there is a strong BIAS risk linked to this. 

6. It is necessary to add a table with the anthropometric data at the beginning of the study of the two groups. 

7. The data in Table 7 would show that coeliac patients eat a healthier breakfast than their non-celiac counterparts. This contradicts the information on coeliac products being higher in sugar and refined flours and sodium. 

8. The discussion should be enriched by arguing the practical aspects of the results and considering the remarks made above. 

Reading the paper is difficult. Too many paragraphs, often unconnected. There are some errors in the English to be fixed. 

Author Response

COMMENTS OF REVIEWER 2
Open Review
Aim of this paper was to assessed ingredients and nutrient composition of gluten-free breakfast products and gluten-containing counterparts and we in a group of Spanish children and adolescents with celiac disease compared to controls.
The paper addresses an interesting issue. In clinical practice it is, in fact, common to have difficulties in the nutritional assessment of the diet in coeliac patients. Especially because comparing breakfast products with gluten and without shows that the latter are often richer in refined flours, sodium, and sugars.
There are some aspects to be improved.
Thank you for reviewing our manuscript. We appreciate the reviewer’s opinion regarding the suitability of the reported manuscript entitled “Is it possible to accomplish a Healthy Breakfast on a Gluten-free Diet? A Study in Children and Adolescents with Celiac Disease in Spain” (Manuscript ID: nutrients-2345610) to be published in Nutrients. As you suggested and in accordance with the other reviewer, English style and grammar has been thoroughly reviewed and general wording also been enriched.

DETAILED COMMENTS

  1. There is plagiarism mainly in the methods part (please check the attached file)
As you have suggested, we have tried reducing the auto-plagiarism, especially in the Participants and Methods Section, but we have also addressed the other reviewer’s comment on trying to provide more information on methodology.
  1. Reading the paper is difficult. Too many paragraphs, often unconnected. 
Considering reviewer´s annotations, we have carefully revised and rewritten some parts of the revised manuscript.
  1. The number of tables is too many. it would be useful to merge them to make it easier to compare the two groups.  (For example, merge table 7 with table 8)
According to your suggestion, we have joined both tables (Page 10).
  1. In the tables, insert the p-value where possible. 
We appreciate the comment, but it is difficult to include other columns in the tables due to space and layout. For this reason, we specified statistically significant data with an asterisk (*) as it indicated in table footnotes.
  1. Such a high figure of adolescents eating breakfast (98%) is astonishing. The figures are much lower. I think there is a strong BIAS risk linked to this. 
Thank you for your comment. The results do indeed indicate higher breakfast consumption data than in other studies, but these were the results reported by the participating volunteers in their dietary surveys. It could be biased, as many other studies, because people volunteering in a scientific research on diets are more probably more healthy-diet conscious. On the other hand, the average age of the participants is 10 years, and the number of adolescents is lower as compared to children. Children and younger adolescents are less willing to skip breakfast as compared to older adolescents.
In any case, our study is observational and aims to compare the quality of breakfast, in terms of nutritional composition and variety of foods proposed by the recommendations, between a population of young coeliacs and controls, which is facilitated by the homogeneity of the sample in this aspect and other characteristics of age and sex, for example. According to our understanding, it would be more biased to make associations between skipping breakfast and some other biomarker, as the first condition is practically non-existent.
  1. It is necessary to add a table with the anthropometric data at the beginning of the study of the two groups. 
We thank you for your comment, but these data are published in Ballestero et al. (2019) “Nutritional Status in Spanish Children and Adolescents with Celiac Disease on a Gluten Free Diet Compared to Non-Celiac Disease Controls”, as they are from the same experimental study previously published, so we considered that it was not necessary to include them again to avoid auto-plagiarism.
  • Ballestero, C.; Varela-Moreiras, G.; Úbeda, N.; Alonso-Aperte, E. Nutritional status in Spanish children and adolescents with celiac disease on a gluten free diet compared to non-celiac disease controls. Nutrients 2019, 11, 2329, https://doi.org/10.3390/nu11102329
  1. The data in Table 7 would show that coeliac patients eat a healthier breakfast than their non-celiac counterparts. This contradicts the information on coeliac products being higher in sugar and refined flours and sodium. 
We would like to thank you for your observation, but these are two different things: in table 7 we show the nutritional composition of the whole breakfast, evaluating all the food groups consumed (cereals, dairy products, fruits, etc.) by volunteers, while in the rest of the tables we compare only commercialized cereal-based products (gluten-free or gluten containing), finding, in this case, the major differences. According to personal election in this type of products, the nutritional quality of breakfast could be different. In any case, we do not consider the celiac breakfast as healthier than their non-celiac participants, i.e., in general, breakfast in the celiac sample contained less saturated fats, but these derived in a great extent from sugars, sweets and confectionery group (with high content in palm and cocoa oil), as compared to controls group, which contributes also to increase other critical nutrients in their diet. On the other hand, the breakfast of celiac volunteers contained a lower protein quantity.
  1. The discussion should be enriched by arguing the practical aspects of the results and considering the remarks made above. 
              Thank you for your comment. The most practical aspect and conclusion of our study is the need for nutritional education in celiac children and teens, as we explained in discussion and conclusions. This study could set the stage for new dietetic guidelines for celiac population which includes an adequate choice of marketed products through the knowledge of ingredients and nutrients. This could be reached also by increasing the food literacy of this population. Furthermore, it is necessary to educate children and adolescents to make healthier choices of carbohydrates and fibre sources that provide less sugars, such as nuts, dried fruits, dates, or peanut pastes.
On the other hand, our results also represent an opportunity for the industry reformulation. GFP have been modified in nutritional composition in the last years, i.e., they include more fibre quantity, but more efforts on sugar and salt contents are especially necessary.
              All these aspects are mentioned at the end of the Discussion Section.

Round 2

Reviewer 1 Report

-        I am not sure if the study is about healthy breakfast or comparing CD and non-CD groups.

-        Line 138 - delete the extra dot.

-        Lines 298 – 300 – It is not gender-matched since previously you mentioned: “A total of 70 participants with celiac disease (CD) (50% females and 80% children) and 67 non-celiac (control) (39% females; 69% children) took part in the survey”

-        Your results should be compared to other studies in the discussion section.

-        Studies on GFP composition and main ingredients should be used in the discussion section.

-        Lines 395 – 397: This information should be used to construct the title and the objective.

-        Line 406 – complete the sentence comparing to the gluten-containing product.

-        Section 5.1 should be inserted before the conclusion, at the end of the discussion section.

-        Conclusion should summarize your main findings comparing both groups.

Thank you for the opportunity to review this manuscript!

Author Response

I am not sure if the study is about healthy breakfast or comparing CD and non-CD groups.

Thank you very much for your comment. Both aspects are indeed part of the objective of the study. On the one hand, the first objective was to assess the nutritional quality (based on ingredients and nutrients) of processed cereal-based products commonly consumed at breakfast, e.g., breads, breakfast cereals, bakery products, etc., both gluten-free products and gluten-containing counterparts. The second objective was to analyze the breakfast quality of a group of children and adolescents with celiac disease compared to a control sample. This analysis was based on the evaluation of the consumption of the different food groups recommended by the Sociedad Española de Nutrición Comunitaria (SENC) (dairy products, cereals, fruits, etc.) as well as the quality of their nutritional composition.

Line 138 - delete the extra dot.

In accordance with your comment, we have corrected this aspect in the last version of the manuscript.

Lines 298 – 300 – It is not gender-matched since previously you mentioned: “A total of 70 participants with celiac disease (CD) (50% females and 80% children) and 67 non-celiac (control) (39% females; 69% children) took part in the survey.”

We would like to thank you for your observation. This study involved age-matched children and adolescents, although it was not possible to achieve a completely gender homogeneous sample population. For this reason, we have removed the gender-matched expression.

Your results should be compared to other studies in the discussion section.

Studies on GFP composition and main ingredients should be used in the discussion section.

Thank you for your recommendations. First, we would like to mention that, up to our knowledge, no other studies report specifically results on nutrient composition from breakfast products in children and adolescents with celiac disease. Nevertheless, and according to your suggestions, we have enriched the discussion with other related studies, in terms of ingredients frequently used in gluten free products, to make a more in-depth analysis of the main ingredients and nutrients of gluten-free processed cereal-based foods and their gluten-containing counterparts. Besides, the results of our study have been compared with other reports dealing with eating habits, the recommended food groups at breakfast to assess the quality breakfast, and its role in health. Please, refer to Discussion Section in the last version of the manuscript.

Lines 395 – 397: This information should be used to construct the title and the objective.

We want to thank you for your comment. We consider that the title, as it stands, includes all your above-mentioned concepts. In fact, we found no comments on the title in your first revision (round 1).

Line 406 – complete the sentence comparing to the gluten-containing product.

As kindly suggested, we have already done so.

Section 5.1 should be inserted before the conclusion, at the end of the discussion section.

Following the reviewer’s suggestion, we have inserted “Section 5.1” before the conclusions as “Section 4.1”.

Conclusion should summarize your main findings comparing both groups.

Thank you for the recommendation, we have already done.

Reviewer 2 Report

The authors clarified some aspects based on my previous remarks. 

Many doubts remain as to why the authors did not do a single paper compared to their previous work https://www.mdpi.com/2072-6643/11/10/2329

This submission seems an unnecessary appendix to the previous one. 

Good

Author Response

The authors clarified some aspects based on my previous remarks.

Many doubts remain as to why the authors did not do a single paper compared to their previous work https://www.mdpi.com/2072-6643/11/10/2329.

This submission seems an unnecessary appendix to the previous one.

Thank you for your comment. Although the two studies are carried out on the same population sample, a group of Spanish children and adolescents with celiac disease, and a similar age population group without the disease (control), the objectives of both studies are different. In the previous article, published in 2019, the aim was to assess nutritional status in celiac children and adolescents, using dietary, anthropometric, and biochemical parameters, as well as assessing bone health and physical activity; to identify specific needs and to evaluate possible deficiencies derived from a long-term gluten free diet. The present article aimed to assess the ingredient and nutrient composition of cereal-based processed gluten-free products consumed in breakfast, in contrast to their gluten-containing counterparts, and the overall nutritional quality of breakfast targeting a Spanish group of children and adolescents with celiac disease comparing to a control group.

We also would like to mention that the reviewers in charge to check our paper published in 2019, indicated that the report was complete and, to a certain extent, a bit extense, thus suggesting not to include more information or more data.

Round 3

Reviewer 1 Report

-        As recommended in the 1st review, you should insert the line number to make our communication easier (it occurs only after page 4).

-        Objective: You should insert the objective as primary and secondary, as you mentioned in response to reviewers “first objective was to assess the nutritional quality (based on ingredients and nutrients) of processed cereal-based products commonly consumed at breakfast, e.g., bread, breakfast cereals, bakery products, etc., both gluten-free products and gluten-containing counterparts. The second objective was to analyze the breakfast quality of a group of children and adolescents with celiac disease compared to a control sample”.

-        “In fact, we found no comments on the title in your first revision (round 1)” - Considering that your article is still under review, even if I didn't mention a point in the first round, there is no impediment for me to make my observations on the manuscript in the other rounds until its approval. Therefore, my recommendation is still the same.

-        Section 2.1: “Current dietary data were obtained in a cross-sectional age and gender-matched” – As I mentioned before, it is not gender-matched since previously you mentioned: “A total of 70 participants with celiac disease (CD) (50% females and 80% children) and 67 non-celiac (control) (39% females; 69% children) took part in the survey”

-        Results: Why did you use median instead mean? In section 2.3 you mentioned using mean, not median, as mentioned in the results section.

-        Insert a dot at the end of the last sentence of page 3.

-        Line 194 – eggs are also widely used in gluten-free bakery and confectionery products to technologically compensate for the withdrawal of gluten. Therefore, eggs in products tend to contribute to their protein content.

Thank you for the opportunity to review this manuscript!

Author Response

As recommended in the 1st review, you should insert the line number to make our communication easier (it occurs only after page 4).

Attending the reviewer’s observation, we have corrected this aspect in the last version of the manuscript. Sorry for the disturbs.

- Objective: You should insert the objective as primary and secondary, as you mentioned in response to reviewers “first objective was to assess the nutritional quality (based on ingredients and nutrients) of processed cereal-based products commonly consumed at breakfast, e.g., bread, breakfast cereals, bakery products, etc., both gluten-free products and gluten-containing counterparts. The second objective was to analyze the breakfast quality of a group of children and adolescents with celiac disease compared to a control sample”.

As kindly suggested, we have already done so (Please, see Page 3, lines 112-122).

- “In fact, we found no comments on the title in your first revision (round 1)” - Considering that your article is still under review, even if I didn't mention a point in the first round, there is no impediment for me to make my observations on the manuscript in the other rounds until its approval. Therefore, my recommendation is still the same.

Following the reviewer’s observation, we have modified the tittle as follows “Nutritional Composition of Breakfast in Children and Adolescents with or without Celiac Disease in Spain. Role of Gluten-free Commercial Products”.

-        Section 2.1: “Current dietary data were obtained in a cross-sectional age and gender-matched” – As I mentioned before, it is not gender-matched since previously you mentioned: “A total of 70 participants with celiac disease (CD) (50% females and 80% children) and 67 non-celiac (control) (39% females; 69% children) took part in the survey”

Taken into account the reviewer’s comment, we have deleted the inexactly information (Please, see Page 2, lines 134).

-        Results: Why did you use median instead mean? In section 2.3 you mentioned using mean, not median, as mentioned in the results section.

Thank you for the puntualisation, we have corrected this mistake (Please, see Page 3, line 192).

-        Insert a dot at the end of the last sentence of page 3.

In accordance with your comment, we have corrected this aspect in the last version of the manuscript.

-        Line 194 – eggs are also widely used in gluten-free bakery and confectionery products to technologically compensate for the withdrawal of gluten. Therefore, eggs in products tend to contribute to their protein content.

According to the reviewer´s observation, we introduce the following information: “Eggs are also widely used in gluten-free bakery and confectionery products to technologically compensate for the withdrawal of gluten, and occasionally in gluten-free breads.”

Thank you for the opportunity to review this manuscript!

Thank you for reviewing our manuscript and for constructive criticism.

Reviewer 2 Report

Thank you for providing additional information regarding the differences between your previous study and the current submission. It's clear that the objectives of both studies are distinct, with the previous study focusing on assessing the nutritional status and health outcomes of children and adolescents with celiac disease, and the current submission focusing on the nutrient composition of breakfast foods consumed by this population.

It's also helpful to know that the reviewers of your previous paper found it to be complete and comprehensive, and suggested not including more information or data. This suggests that your team carefully considered the scope and focus of each study, and that the current submission is a necessary and valuable addition to the literature on celiac disease and nutrition.

Thank you for your response, and best of luck with your research.

OK

Author Response

Thank you for providing additional information regarding the differences between your previous study and the current submission. It's clear that the objectives of both studies are distinct, with the previous study focusing on assessing the nutritional status and health outcomes of children and adolescents with celiac disease, and the current submission focusing on the nutrient composition of breakfast foods consumed by this population.

It's also helpful to know that the reviewers of your previous paper found it to be complete and comprehensive, and suggested not including more information or data. This suggests that your team carefully considered the scope and focus of each study, and that the current submission is a necessary and valuable addition to the literature on celiac disease and nutrition.

Thank you for your response, and best of luck with your research.

Thank you for reviewing our manuscript and for constructive criticism. We appreciate your opinion about the suitability of the revised manuscript for being published in Nutrients.
